# Single-fed broadband CPW-fed circularly polarized implantable antenna for sensing medical applications

**Arslan Dawood Butt**[1☯], **Jalal Khan**[2☯], **Sarosh Ahmad**[1,3☯]*, **Adnan Ghaffar**[4‡], **Ahmed Jamal Abdullah Al-Gburi**[5‡], **Mousa Hussein**[6]*

**1** Department of Electrical Engineering and Technology, Government College University Faisalabad (GCUF), Punjab, Pakistan, **2** Department of Telecommunication Engineering, University of Engineering and Technology, Mardan, Pakistan, **3** Department of Signal Theory and Communications, Universidad Carlos III de Madrid (UC3M), Leganes, Madrid, Spain, **4** Department of Electrical and Electronic Engineering Auckland University of Technology, Auckland, New Zealand, **5** Department of Electronics and Computer Engineering (FKEKK), Center for Telecommunication Research and Innovation (CeTRI), Universiti Teknikal Malaysia Melaka (UTeM), Durian Tungal, Malaysia, **6** Department of Electrical and Communication Engineering, United Arab Emirates University, Abu Dhabi, UAE

☯ These authors contributed equally to this work.
‡ AG and AJAA also contributed equally to this work.
* saroshahmad@ieee.org (SA); mihussein@uaeu.ac.ae (MH)

**Data Availability Statement:** All data files are available from https://doi.org/10.6084/m9.figshare.21579753.v1.

## Abstract

Biomedical telemetry relies heavily on implantable antennas. Due to this, we have designed and tested a compact, a circularly polarized, a low-profile biomedical implantable antenna that operate in the 2.45 GHz ISM band. In order to keep the antenna compact, modified co-planar waveguide (CPW) technology is used. Slotted rectangular patch with one 45-degree angle slot and truncated little patch on the left end of the ground plane generate a frequency-range antenna with circular polarization. Using a 0.25-millimeter-thick Roger Duroid-RT5880 substrate with a thickness of εr = 2.2, tanδ = 0.0009 provides flexibility. The volume of the antenna is 21 mm x 13.5 mm x 0.254 mm (0.25λg × 0.16λg × 0.003λg). The antenna covers 2.35–2.55 GHz (200 MHz) in free space and 1.63–1.17 GHz (1.17 GHz) in epidermal tissue. With skin tissue that has more bandwidth, the (x and y)-axis bends of the antenna are also simulated via the simulation. Bended antenna simulations and measurements show excellent agreement. At 2.45 GHz, the skin-like gel had -10dB impedance and 3dB axial ratio (AR) bandwidths of 47.7 and 53.8%, respectively. The ultimate result is that the SAR values are 0.78 W/kg in skin over 1 g of bulk tissue, as determined by simulations. The suggested SAR values are lower than the FCC's maximum allowable limit (FCC). This antenna is small enough to be implanted in the body, making it perfect for biomedical applications.

## Introduction

An ever-increasing number of people are benefiting from implantable medical devices (IMDs) [1, 2]. Using an implanted antenna for radio networks among IMDs and external monitoring

**Competing interests:** The authors have declared that no competing interests exist.

**Fig 1. The implantable medical devices schematic diagram (IMDs) [3].**

equipment, the telemetry system has provided various benefits, including higher data rates and longer connection distances [3]. Implantable medical devices (IMDs) are critical to the proposed CP antenna concept [4]. There must be a very low SAR and a highly flexible antenna for human care applications in the in-body antennas [5]. Therefore, the antenna must be consistent and malleable with the human body throughout the design phases. In order to avoid damaging human tissues, the antenna should have a wider bandwidth that can span the needed frequency spectrum once it has been tested on human tissues. The CP antenna's radiation qualities frequently fluctuate owing to the human body's many layers structure. The human body consists of skin, fat, bone, muscle, and blood. A biocompatible CP antenna that can be used on the human body is challenging to construct since the tissues have varying dielectric characteristics. Many in-body CP antennas are created for wireless communication between (402–405 MHz) [6–12] and (2.4–2.48 GHz) [13–18] in the mid radio band range. Biomedical implants may now make use of a small, broadband antenna operating in the MICS band (403 MHz). With a thickness of 1.27 mm, the Roger 6010 semi-flexible substrate material was employed. Compared to our proposed design, SAR values were calculated to be 284.5 W/kg over 1g. Another Polyimide-based implantable biomedical antenna operating in the ISM band was recently disclosed [19, 20]. Fig 1 depicts a schematic representation of implanted medical devices (IMDs).

The antenna's achieved gain was -16.8 dB and its operating bandwidth was 1.22%. The antenna has a total surface size of 25x20 mm$^2$. The actual gain and operational bandwidth were found to be -34.9 dB and 14.9% across the specified frequency range. For medical applications, an improved impedance matching circularly polarized antenna with a compact size was reported [21]. The ground plane features an X-shaped slot and a circular radiating patch. As the substrate material, Rogers 3010 with a substrate height of 0.634 mm was used. Due of the double-layered substrate, the antenna has a thick profile. In contrast to the IEEE/IEC 6270–1 limitation of 649 W/kg for 1g of bulk tissue, the antenna had a SAR value of around 649 W/kg and was less biocompatible with the human body. Utilizing an annular-ring-shaped, circular CP antenna with a 19.1% axial ratio bandwidth may be advantageous for biomedical implants.

In addition to the two layers of flexible Roger 3010 substrates with a total layer height of 0.63 mm, this circular antenna had a two-layer structure. In the ISM band, the antenna operated at 2.45 GHz with an 8% bandwidth. This antenna's SAR values were discovered to be 508 W/kg, which exceeds the standard limit. The maximum gain of the antenna is -17.5 dB. The antenna was larger than our suggested implantable flexible antenna, and no bending testing was conducted. An implantable antenna based on a split-ring resonator (SRR) supplied by CPW was shown in another work [22–24]. The antenna had a total surface area of 24 x 22 mm$^2$ and a flexible Polyimide substrate with a thickness of 0.07 mm. In-vitro testing is ideal given the antenna's modest profile. The antenna's size is still somewhat enormous compared to the antenna that we had planned on designing [25–27]. At 2.45 GHz, the antenna's actual gain was -19.7 dB, and its operating bandwidth was 24.4%. Table 1 shows a comparison of the proposed antenna's performance with that of current studies.

For biomedical applications, this article discusses a circularly polarized flexible implantable antenna operating at 2.45 GHz in the ISM band. The antenna's dimensions are $221 \times 13.5 \times 0.254$ mm$^3$ ($0.25\lambda g \times 0.16\lambda g \times 0.003\lambda g$). RT5880 flexible substrate ($\varepsilon r = 2.2$, $\tan\delta = 0.0009$) is used in the design. There are two types of radiation patterns for the antenna: elliptical (E-plane) and omnidirectional (H-plane). The compact size and flexibility of the implanted antenna improves its function in the human body. CPW technology is used in the design of the stated antenna, resulting in improved bandwidth response and decreased antenna back-radiation. These physical layers, such as the outer layer's dielectric

**Table 1. Performance comparison of the proposed antenna with recent work.**

| Ref. No. | Dimensions (mm$^3$) / ($\lambda g^3$) | Operating Frequency (GHz) | Substrate Material ($\varepsilon_r$) | Realized Gain (dB) | Imp. BW / AR BW (%) | Polarization/ Antenna Type | Human Skin Tissue | SAR (W/kg) @ 1g |
|---|---|---|---|---|---|---|---|---|
| [7] | (22.5x22.5x2.5) (0.59x0.59x0.065) | 0.403 & 2.45 | Roger 3210 (10.2) | -26 / -15 | 35.3&7.1 | Linear/ Zig-Zag Patch | SKIN | Not Calculated |
| [8] | (23x16.4x1.27) (0.098x0.070x0.005) | 0.402 | Roger 6010 (10.2) | -34.9 | 12.9 | Linear/ PIFA | SKIN | 284.5 |
| [9] | (22x23x1.27) (0.57x0.6x0.033) | 0.402 & 2.4 | Roger 3010 (10.2) | -36.7 / -27.1 | 7.4 & 6.6 | Linear / Slotted Patch | SKIN | 832/ 690 |
| [10] | (25x20x0.07) (0.34x0.27x0.00095) | 2.45 | Polyimide (2.78) | -16.8 | 1.22 | Linear/ Printed Patch | SKIN | 1.0 |
| [11] | (15x21.5x1.5748) (0.39x0.56x0.039) | 2.45 | Roger 3210 (10.2) | <-15 | 3.2 | Linear/ Square Patch | SKIN | Not Calculated |
| [12] | (3.14x100x2.54) (0.082x2.6x0.066) | 0.4/2.45 | Roger 6010 (10.2) | -33.1/ -14.55 | 38.1 & 17.6 | Linear/ Circular Patch | SKIN | 241.5/ 149.7 |
| [13] | (10.4x10.4x0.508) (0.145x0.145x0.007) | 2.45 | Taconic (2.95) | -34 | 16.3 / 15.1 | Circular/ Printed Monopole | SKIN | 356.4 |
| [14] | (14x14x0.5) (0.365x0.365x0.013) | 2.45 | Roger 3010 (10.2) | -15.96 | 18.36 / 6.93 | Circular/ Printed Monopole | SKIN | 0.494 |
| [15] | (40x40x1.59) (1.37x1.37xo.054) | 1.76/488 | FR-4 (44) | -39/ -22 | 20.3 & 32.11 | Linear/ Monopole | SKIN | Not Calculated |
| [16] | (3.14x286x1.34) (0.082x0.748x0.035) | 0.402/2.45 | Roger 3010 (10.2) | -41 / -21.3 | 41 & 27.8 | Linear/ Circular PIFA | SKIN | 666/676 |
| [17] | (3.14x23x0.634) (0.082x0.60x0.016) | 2.45 | Roger 3010 (10.2) | -20.3 | 16/ 18.3 | Circular/Slotted Circular Patch | MUSCLE | 649 |
| [18] | 3.14x25x1.27) (0.082x0.65x0.033) | 2.45 | Roger 3010 (10.2) | -17.5 | 8 / 19.1 | Circular / Annular Ring | MUSCLE | 508 |
| [19] | (24x22xo.07) (0.36xo.33xo.001) | 2.45 | Polyimide (3.5) | -19.7 | 24.4 | Linear / Monopole | SKIN | 0.719/ 0.229 |
| [This work] | (21x13.5x0.254) (0.25x0.16xo.001) | 2.45 | Roger RT5880 (2.2) | -15.8 | 47.7 / 53.8 | Circular /Simple Monopole | SKIN | 0.78 |

characteristics and electrical conductivity as well as the weight of the antenna, were used to create and examine this tiny antenna. The antenna's parameters such as SAR and realized gain were also determined in this article. As a substrate, it's made of a flexible substance that's capable of being bent. It is also important to keep in mind that antenna bending will have a negligible effect on performance when it is evaluated in human tissues. Tests of the antenna's performance are conducted in close proximity to human flesh. For example, the skin phantom's surface area is restricted to 50x50 mm$^2$. For 2.45 GHz, this antenna's peak gain is equivalent to that of the prior study, thanks to the new redesigned patch as well as its CPW. SAR values have been greatly lowered as a result of improvements in antenna efficiency and performance under the effect of the human body [28–30]. This article begins with an overview of the research project, which includes a summary of the issue, the study aims, and the research's importance. Design analysis and manufacturing and measurements results as well as testing in skin tissue are also included in this section. The conclusion is also included at the end.

## Proposed design method

One of the most compact printed monopole antennas has been presented, which is fed via a modified co-planar waveguide. There are three separate layers in the antenna depicted in Fig 2, which are the ground plane (CPW), the low-profile flexible Roger RT5880 substrate ($\varepsilon_r$ = 2.2 and tan$\delta$ = 0.0009), and the printed monopole (Fig 2). $21 \times 13.5 \times 0.254$ mm$^3$ ($0.24\lambda g \times 0.16\lambda g \times 0.003\lambda g$). is the antenna's total volume. Using CST software, the antenna's $S_{11}$ is determined to be -29.8dB at the operating frequency of 2.45 GHz, which is consistent with the stated antenna. The antenna's operational bandwidth and axial ratio bandwidth seem to be 5.73% and 47.3%, respectively. Table 2 summarizes the antenna's ideal performance parameters.

Here is a breakdown of the design process for the implanted antenna:

The basic antenna design consists of a 50-ohm CPW feedline and a printed rectangular monopole (ANT I in Fig 3A). Using (1) and (2) [31, 32], the width and length of the monopole are determined:

$$Wm = \frac{\lambda_o}{2(\sqrt{0.5(\varepsilon_r + 1)}}$$

(1)

At the operating frequency, $\varepsilon_r$ and $\lambda_o$ are the substrate's relative permittivity and wavelength in free space. A good choice of 'Wm' results in a perfectly matched impedance. In order to get the monopole's length, we may use Eq (2):

$$Lm = \frac{c_o}{2f_o\sqrt{\varepsilon_{eff}}} - 2\Delta L_m$$

(2)

Here, Wm and Lm represent the width and length of the antenna in millimeters. Also included are the speed of light, the change in the antenna's length due to its fringe effect, and its effective dielectric constant. Utilizing Eq (3), one can determine the effective relative permittivity.

$$\varepsilon_{eff} = \frac{\varepsilon_r + 1}{2} + \frac{\varepsilon_r - 1}{2}\left(\frac{1}{\sqrt{1 + 12\frac{ts}{W_m}}}\right)$$

(3)

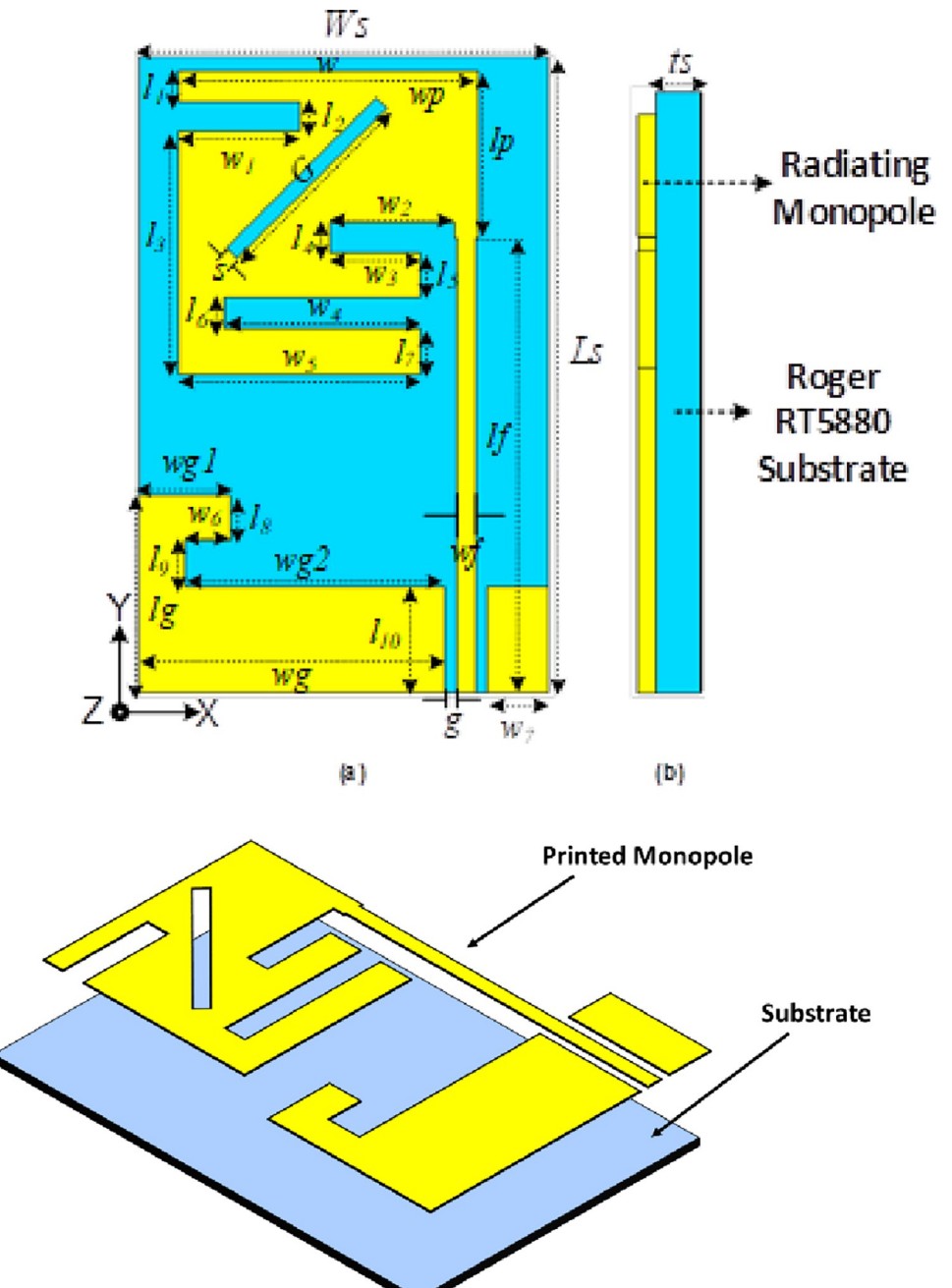

**Fig 2.** Antenna proposal with dimensions, (a) front view, (b) side view, and (c) 3D perspective.

'ts' is the substrate thickness that is used in this equation. Using Eq (4), the fringing effect may be calculated.

$$\Delta Lm = 0.421 t_s \frac{(\varepsilon_{eff} + 0.300)\left(\frac{W_m}{t_s} + 0.264\right)}{(\varepsilon_{eff} - 0.258)\left(\frac{W_m}{t_s} + 0.813\right)} \tag{4}$$

The rectangular printed monopole's starting parameters are 'Lm' and 'Wm,' with εr = 2.2,

**Table 2. List of optimized parameters for the proposed antenna.**

| Parameters | Values (mm) | Parameters | Values (mm) | Parameters | Values (mm) |
|---|---|---|---|---|---|
| $l_s$ | 21 | $l_8$ | 1.5 | w2 | 4.1 |
| $l_1$ | 1.0 | $l_9$ | 1.5 | w3 | 3.0 |
| $l_2$ | 1.0 | $l_{10}$ | 5.0 | w4 | 6.5 |
| $l_3$ | 8.0 | g | 0.4 | w5 | 8.0 |
| $l_4$ | 1.0 | lf | 15 | w6 | 1.5 |
| $l_5$ | 1.0 | wf | 0.6 | w7 | 2.0 |
| $l_6$ | 1.0 | ws | 13.5 | wg | 10.1 |
| $l_7$ | 1.5 | $w_1$ | 4.0 | wg2 | 8.0 |
| lg | 8.0 | wgl | 3.0 | wp | 9.9 |
| G | 7.0 | S | 0.5 | ts | 0.254 |
| $l_p$ | 5.5 | w | 9.9 | | |

ts = 0.254 mm in (1)-(4). The patch's width and length are Lm = 10 mm and Wm = 9.9 mm after complete wave optimization.

## Stated design procedure

In order to get the most out of the antenna design. The antenna must be divided into stages so that the primary difference in results may be plainly seen. Therefore, in this part, we have included the antenna design stages provided. Fig 3 depicts the many stages in the development of the printed monopole antenna. When designing antenna ANT I, we used the basic rectangular form with the 10dB reflection coefficient but it didn't show up in the target band and just resonated at 3 GHz without exhibiting any circular polarization, therefore we had to change the design. "Lm" and "Wm" are the monopole's dimensions, which are 10 millimeters in length and 9.9 millimeters in width. In order to move the antenna's frequency from 3 GHz to the necessary 2.45 GHz band and achieve circular polarization, the top and bottom sides of a rectangular monopole are slotted as described in ANT II (see Fig 3(B)). Resonating frequency was moved from 3 to 2.7 gigahertz, and the S11 value was maintained at or below -10 dB at 2.55 gigahertz, with an overall bandwidth of 5.3%. As a result, the frequency is moved by 100 MHz (from 2.7 GHz to 2.6 GHz) in the third example, while the circular-polarization behavior is seen in the second case at 2.55 GHz. It's for this reason that we have a 45-degree horizontal slot at the ground plane's edge, and a brief truncated patch that truncates the ground plane's frequency range, as can be seen in ANT IV. For example, in this situation, the impedance bandwidth of the printed monopole antenna at 2.45 GHz is 47.3%, and it is performing flawlessly at this frequency. Fig 3(E) compares the simulated $S_{11}$ of all the design processes and shows how simple it is to tune impedance at 2.45 GHz using the CPW approach. Fig 3(F) depicts a comparison of axial ratios.

## Optimization of parameters of the antenna

This section delves further into the investigation of the parameters of the implanted antenna. Changing the values of critical parameters such as the width of the monopole "W1," the width of the feedline "wf," the width of the CPW ground plane "Wg," the length of the monopole "l2," and the length of the horizontally rotated slot "G" can be used to adjust the reflection coefficient at 2.45 GHz, as shown in Fig 4. 4th Fig When the value of 'W1' is reduced from 4 to 6 millimeters, the frequency range shifts to 2.45 GHz, as illustrated in Fig 4A (150 MHz). As seen in Fig 4(B), altering the value of 'Wf' from 7 to 9 mm results in a frequency band shift from 2.2 to

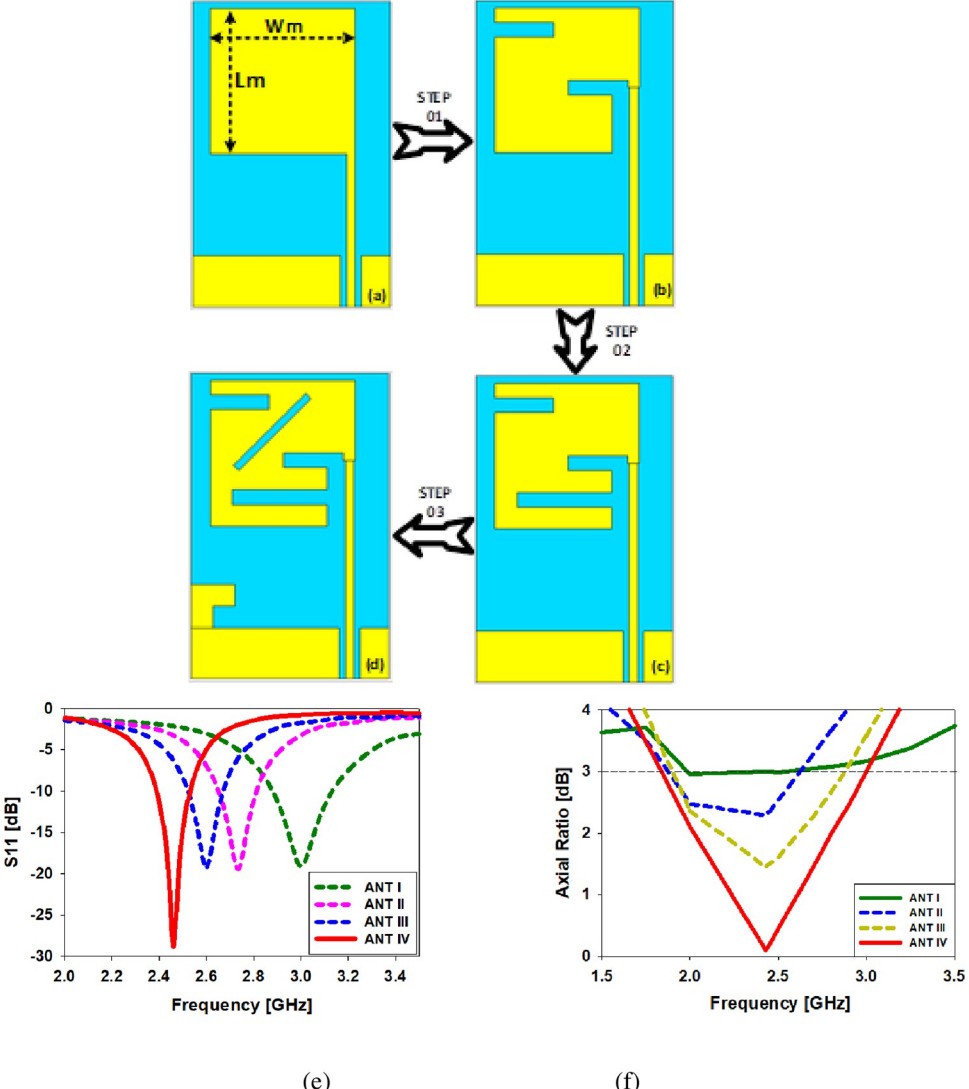

**Fig 3.** The antenna design process begins with the rectangular monopole, followed by a slotted monopole, a modified slotted monopole, and a modified slotted monopole with a ground plane (ANT IV), (e) S11 simulation findings, (f) axial ratio comparison in dB.

2.63 GHz. Fig 4(C) illustrates the frequency range fell from 2.45 GHz to 2.39 GHz as the 'Wg' values grew from 1 millimeter to 3 millimeters (60 MHz). When 'l2' is extended from 0.254 mm to 1 mm, the frequency shift in Fig 4 shifts from 2.45 GHz to 2.6 GHz, a 150 MHz difference (d). As shown in Fig 4(E), raising the value of the 'lg' parameter from 7 mm to 9 mm resulted in a band shift from 2.45 GHz to 2.82 GHz (370 MHz)). The frequency was increased from 2.8 gigahertz to 2.4 gigahertz (400 MHz; see Fig 4(F)) by raising the value of the slot 'G' from 4 millimeters to 6 millimeters.

## Fabricated prototype of the stated antenna

As seen in Fig 5, a 0.254-millimeter-thick flexible Roger RT5880 substrate has been used to build the biomedical implantable monopole antenna. Fig 5(C) depicts the configuration used to examine the radiation pattern in an anechoic chamber. Radiofrequency (RF) absorbers are

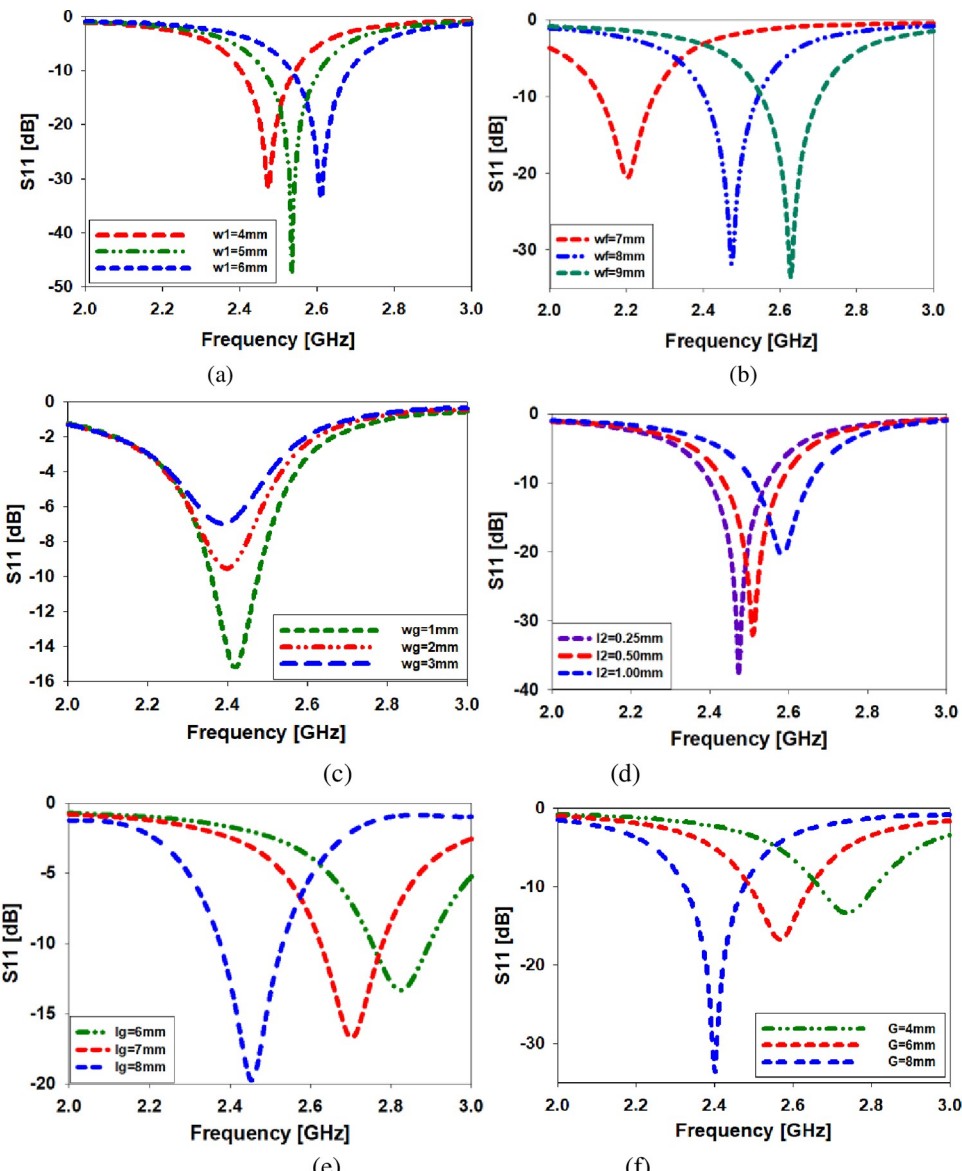

**Fig 4. Variations in "w1," "wf," "Wg," "L2," "LG," and "G" are all studied parametrically to see how they affect the antenna's performance.**

utilized in the farfield assessment chamber to absorb RF signals from the proposed printed antenna within the chamber. Checking the implanted antenna's S11 requires the use of a vector network analyzer. Antenna design for 2.39–2.53 GHz at 2.45 GHz covers 140 MHz of bandwidth in the simulated results, whereas antenna design for 2.45–2.55 GHz at 2.45 GHz has a measurement of 200 MHz of bandwidth (Fig 6).

(E & H)-plane antenna simulation and open space measurement (see Fig 7). The antenna's behavior may be described using a 2D radiation pattern. On E-plane the antenna produces an elliptical pattern, whereas on H plane the radiation pattern produces an omnidirectional pattern. At 2.45 GHz, the designed antenna has a modelled and observed gain of -2.5 dB and -2.7 dB, respectively. Fig 7 displays the implanted antenna's current density (b). The CPW feedline, ground plane, and lower slots are the primary conduits for current flow. To demonstrate

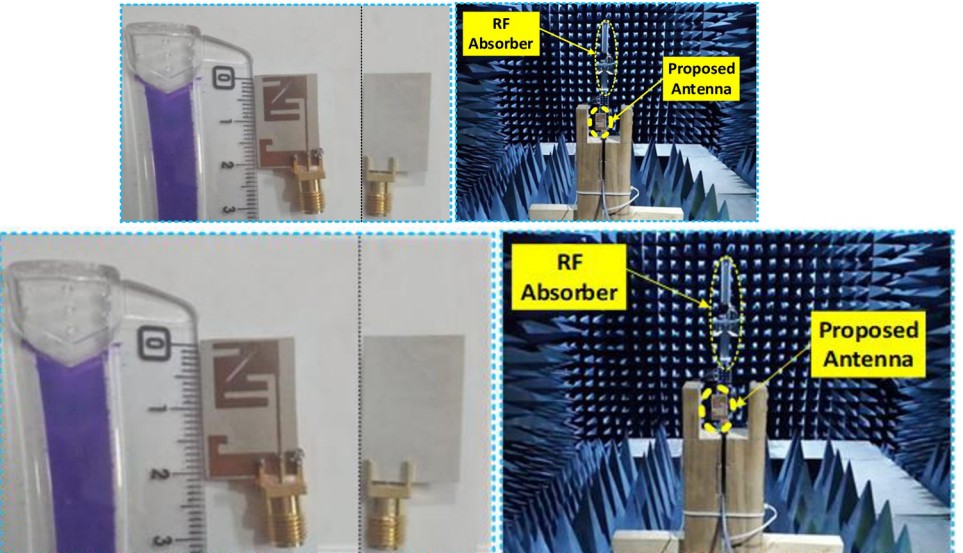

**Fig 5. A top view, a bottom view, and a configuration of the antenna within an anechoic chamber are shown for the prototype antenna that was manufactured.**

circular polarization, the current flows around the printed monopole in a round pattern. Fig 8 illustrates the contrast of the simulated and observed axial ratios and the peak gain graph. An antenna's axial ratio bandwidth in free space can be clearly seen from Table 3 and the graph, which shows that while the simulation antenna covers 47.3% of this bandwidth in free space,

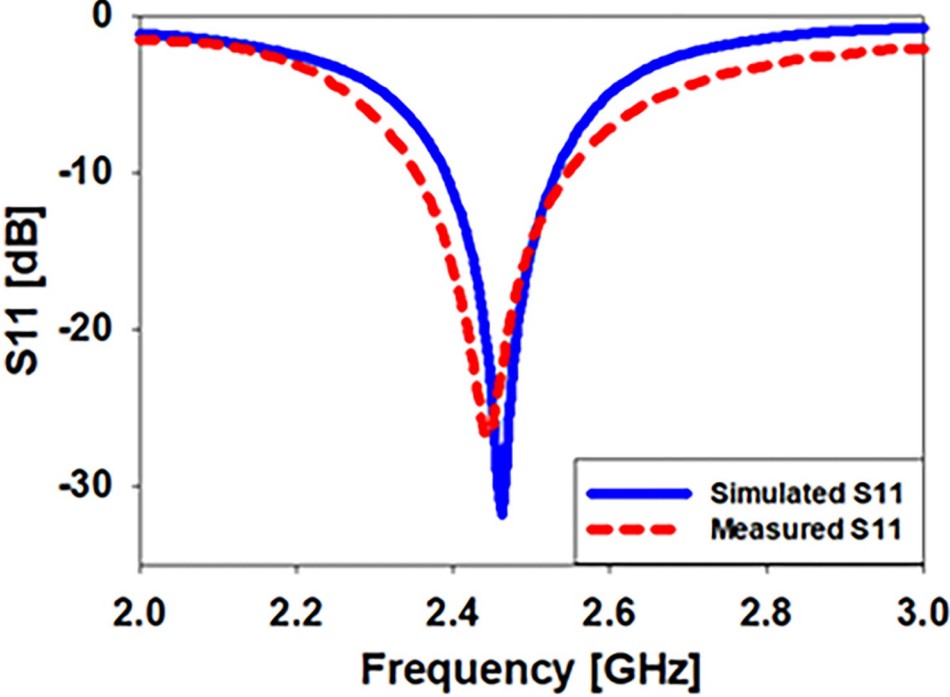

**Fig 6. Comparison of the printed prototype antenna's modelling and measurement of its free space reflection coefficients.**

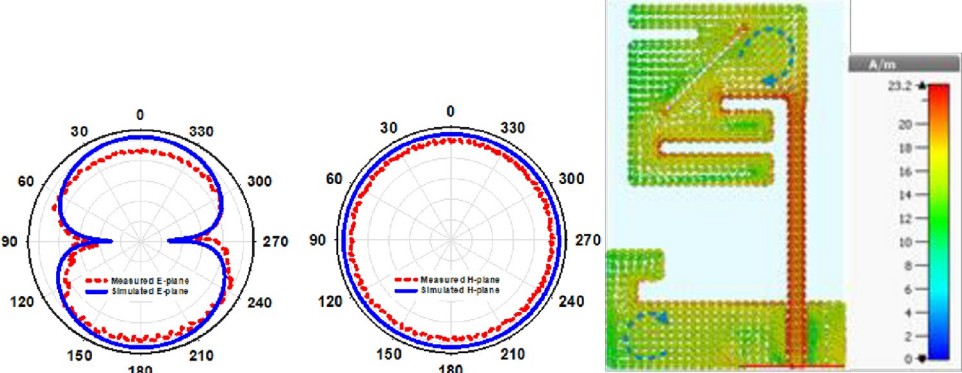

**Fig 7.** (a) 2.45 GHz antenna open space 2D radiation pattern, (b) 2.45 GHz radiating monopole current distribution.

the actual antenna only covers 35.9% of it at the same frequency at 2.45 GHz, as shown in Table 3.

## Analysis of the antenna's bending behavior when it is flexible

In order to verify the antenna's $S_{11}$, it is required to do a bending study along the horizontal and vertical axes of the antenna, which is made up of a flexible Roger RT5880 substrate. The twisting of the antenna in implanted situations is critical. Antenna bend analysis in free space along the x- and y-axes is described in this section. Using a cylindrical foam with a diameter of 30 mm, the bending analysis was both simulated and empirically confirmed.

**Bending along X-axis.** When it comes to studying the S11 behavior, peak gain, and far-field of an antenna, the antenna's twisting radius (Bx = 20 mm to 100 mm) has been picked to investigate the antenna's S11 behavior, peak gain, and farfield. As shown in Fig 9(A), when the antenna is bent along the x-axis, the simulated results are compared. It is well-known that the radiation pattern at 2.45 GHz has a sliding loss in gain due to needless modifications in all twisting radii. When comparing antenna axial ratios, Fig 9(B) illustrates this contrast. The graph shows that the axial ratio's values change somewhat when bent along the x-axis. It has an axial ratio of less than 3dB and is completely resonant at a frequency of 2.45 GHz.

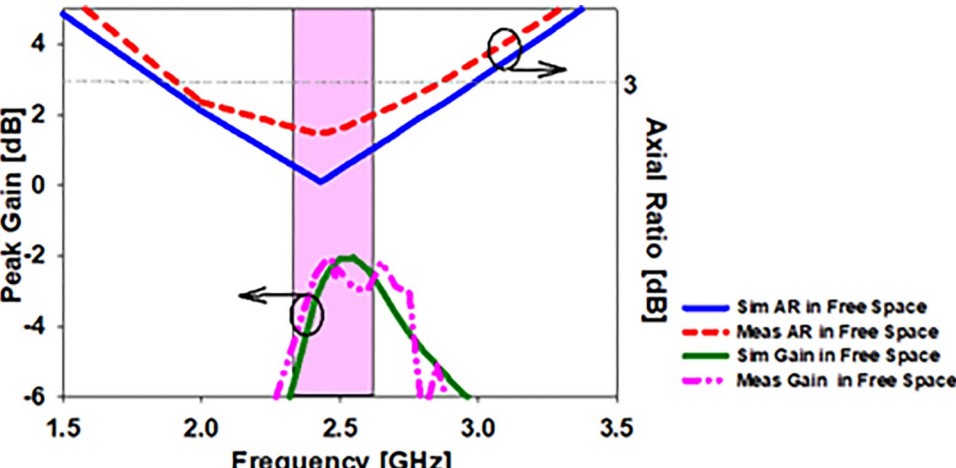

**Fig 8. The gain and axial ratio of the printed monopole antenna when it was tested in open space, as well as the results of the simulation.**

**Table 3. A comparison of the simulated and observed findings was carried out in free space at 2.45 GHz.**

| Simulated in Free Space | | | Measured in Free Space | | |
|---|---|---|---|---|---|
| Gain (dB) | Imp. BW (%) | A.R BW (%) | Gain (DB) | Imp. B.W (%) | A.R B.W (%) |
| -2.5 | 5.71 | 47.3 | -2.7 | 8.1 | 35.9 |

Using computer simulations and actual measurements, the return loss of the stated biomedical monopole printed antenna was calculated. As shown in Fig 10 (Bx = 30mm), the simulated and observed $S_{11}$ values are compared. $S_{11}$ is simulated to cover the bandwidths from 2.38–2.53 GHz, whereas the antenna contains the bandwidths from 2.43–2.49 GHz at 2.45 GHz in real-world $S_{11}$.

Fig 11 depicts the radiation pattern along (E & H)-plane in free space when the flexible antenna is bent along the x-axis, and the antenna's 2D radiation pattern shows how it works in terms of radiation. The antenna described has an elliptical pattern and omnidirectional radiation on the E-plane and H-plane, respectively. We discovered the antenna's simulated peak gain was -2.14 dB and its observed peak gain was -2.3 dB when doing an x-axis bending analysis at Bx = 30mm.

**Bending along Y-axis.** Measurement of twisting radii along the y-axis may verify $S_{11}$ behavior, true gain, and antenna radiation pattern (bending from 20 mm to 100 mm). When the antenna is bent in a y-direction, the simulation results are shown in Fig 12(A). The radiation pattern at 2.45 GHz displays a considerable loss in gain for all twisted radii. Fig 12(B) shows the axial ratio of the bended antenna on the y-axis in contrast to what is shown in Fig 12 (A). The axial ratio values shift somewhat as the y-axis is bent in this graph. Its axial ratio is less than 3 dB, and it works well at 2.45 GHz.

When bent in the y-axis, the $S_{11}$ of the proposed flexible implantable antenna is simulated and measured. $S_{11}$ (By = 30mm) is shown in Fig 13 as a comparison between the simulation and the observed $S_{11}$. As predicted, the bent antenna covers a range of 2.39–2.52 GHz (130MHz) when measured at 2.45 GHz, whereas in the case of simulation, it covers a range of 2.40–2.49 GHz (90MHz) when measured at the same frequency.

Fig 14 depicts the radiation pattern in free space when the flexible antenna is bent along the y-axis as a result of modelling and observations. The antenna's 2D radiation pattern explains how it works in terms of radiation. E-plane pattern is elliptical whereas H-plane radiation characteristics are omnidirectional in the 2.45 GHz frequency spectrum. As we were bending the y-axis, we saw that the antenna's peak gain was 2.12 dB, although the perceived peak gain was 2.35 dB.

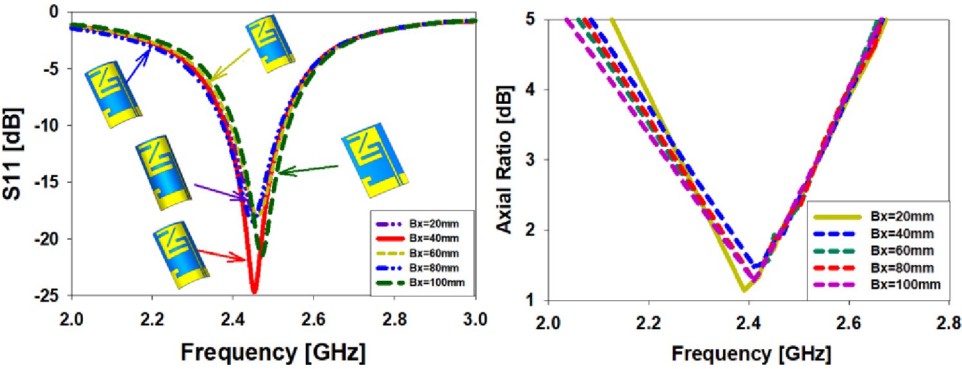

**Fig 9.** Antenna bending analysis at 2.45 GHz, (a) difference in $S_{11}$, (b) comparison in AR.

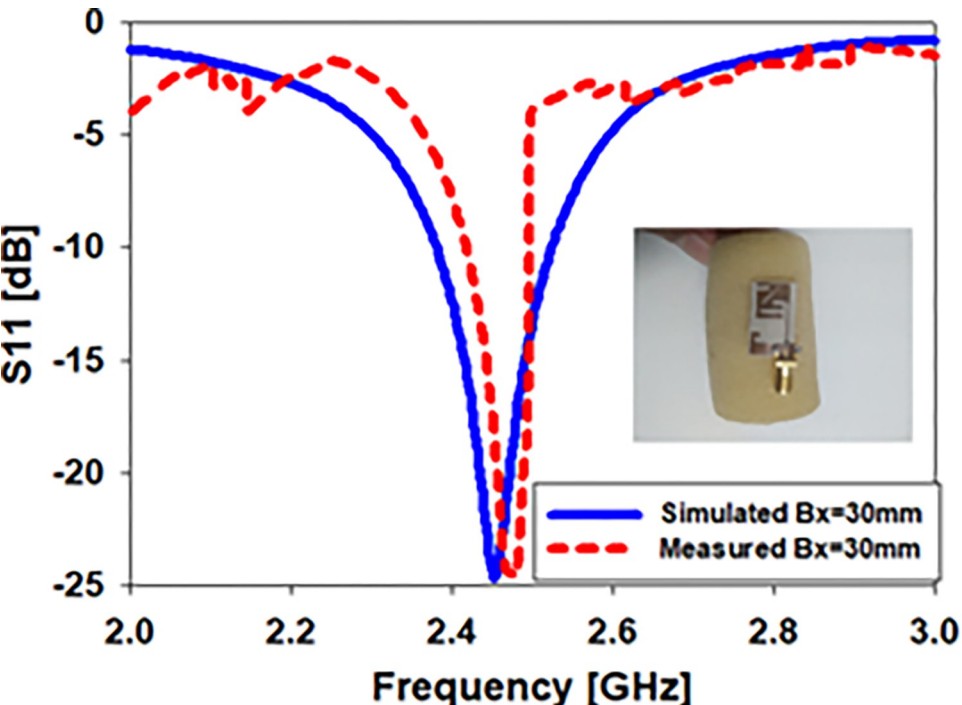

**Fig 10. The antenna was bent along the x-axis by 30 millimeters, and its $S_{11}$ response was observed after the bend.**

## Testing inside skin

In this part, the suggested design is then evaluated on a phantom human body to investigate the antenna's suitability for biomedical telemetry applications. SAR calculations must take into account the human skin phantom in order to build an antenna with high accuracy for biotelemetry applications. A human skin phantom is used to test antenna performance and SAR values at 2.45 GHz. Fig 15 shows that the skin phantom box has a 50 x 50 mm$^2$ area. A simulation of the stated design is carried out using the CST program. An ideal situation would be to design in-body antennas that prevent major coupling effects from human tissue. There are

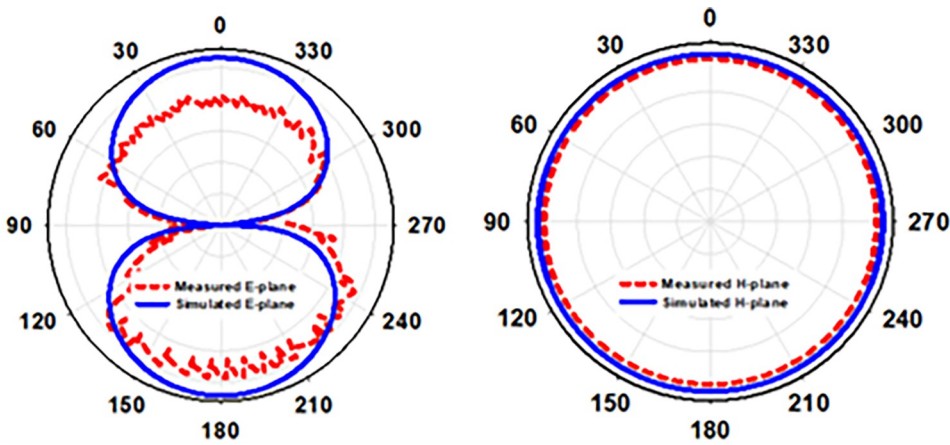

**Fig 11. Radiation pattern of an antenna in free space when it is bent in two dimensions along the x-axis (Bx = 30mm).**

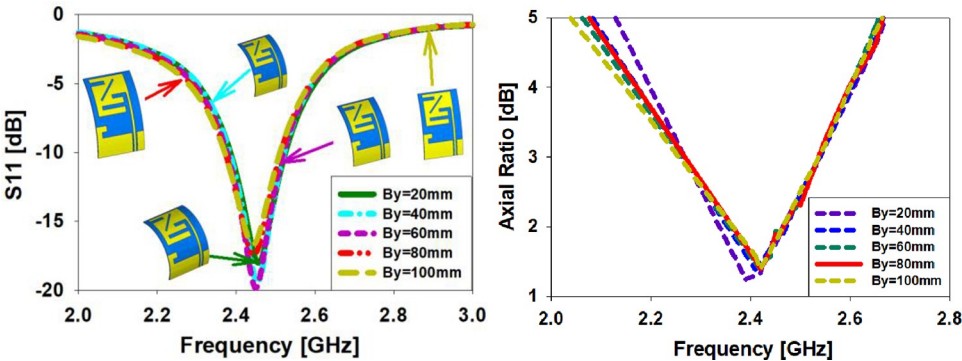

**Fig 12.** S11 and AR antenna bending at 2.45 GHz are compared in (a) and (b) respectively.

many similarities between skin and 'Skin-Mimicking Gel' because of its ɛr of 46 (see Fig 16). This is used to evaluate the radiation pattern and scattering properties of the implantable printed monopole antenna in Skin. To prepare a solution of the "skin-like gel", the amount and percentage of the ingredients of the gel such as sucrose (53%), deionized water (47%), and sodium chloride (NaCl) solution are adjusted to 40 ml in a beaker with a magnetic stirrer and stimulated for 15 min. Then, 0.5 g of dry carbomer was added in a liquid solution to form a 'skin like gel'. The solution was then heated to 81˚C for 59 min to obtain a neat mixture. The solutions were then coagulated by chilling to room temperature before being measured [33–35].

Fig 17 depicts the difference among the experimental and measurement results of $S_{11}$ of the antenna within the skin. According to the simulated S11 within the skin, the biomedical antenna operated between 1.6 and 2.75 GHz (46.7%) at 2.45 GHz whereas the constructed

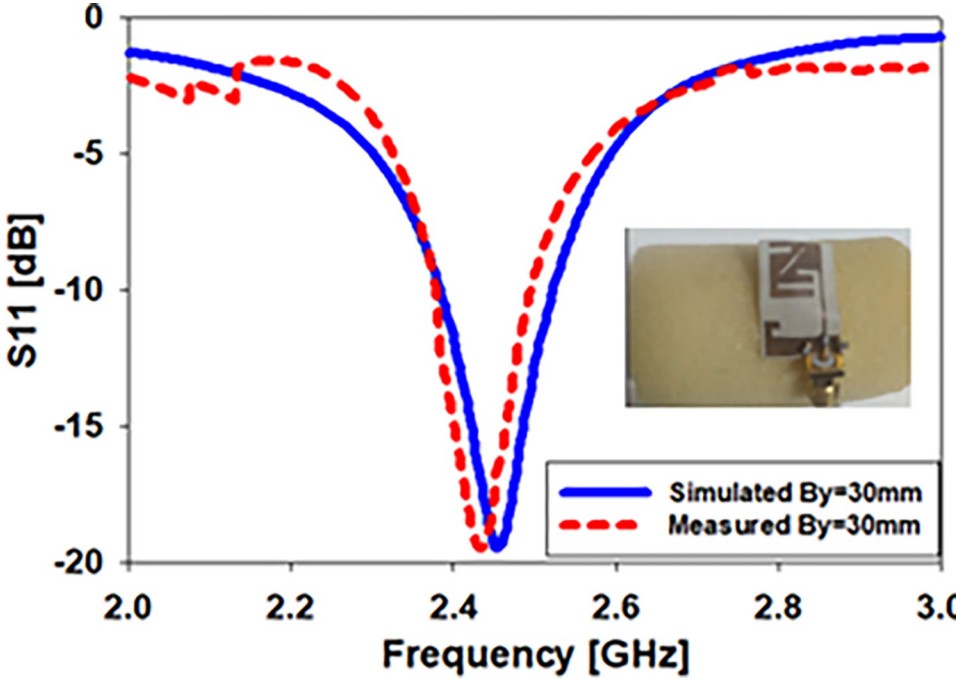

**Fig 13. Flexibility and S11 behavior in a y-axis bent antenna (By = 30 mm).**

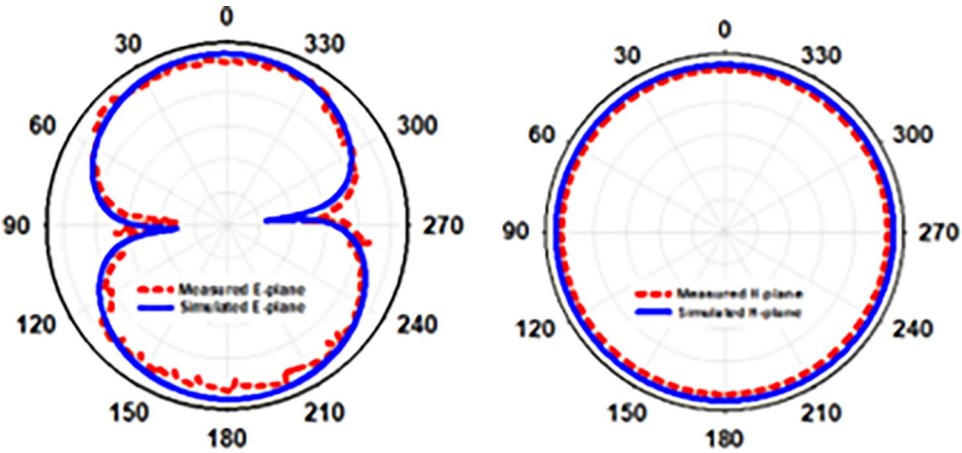

**Fig 14. Radiation pattern of antenna in free space when bent along y-axis (By = 30mm).**

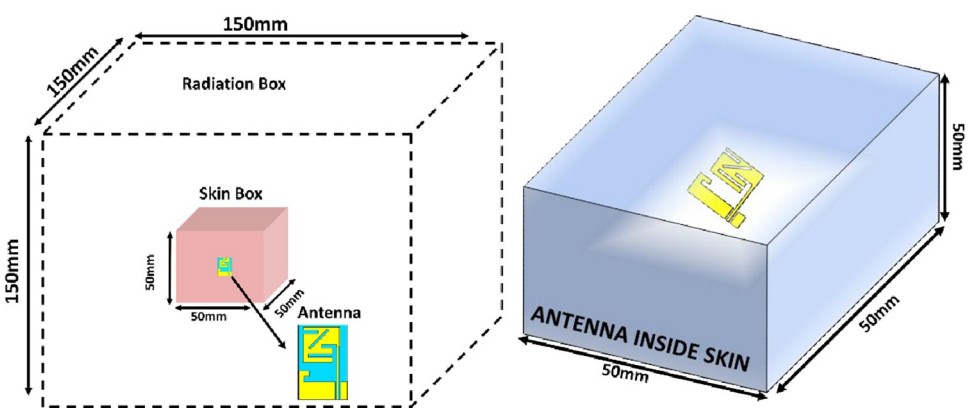

**Fig 15. Perspective and side views of a human skin phantom box.**

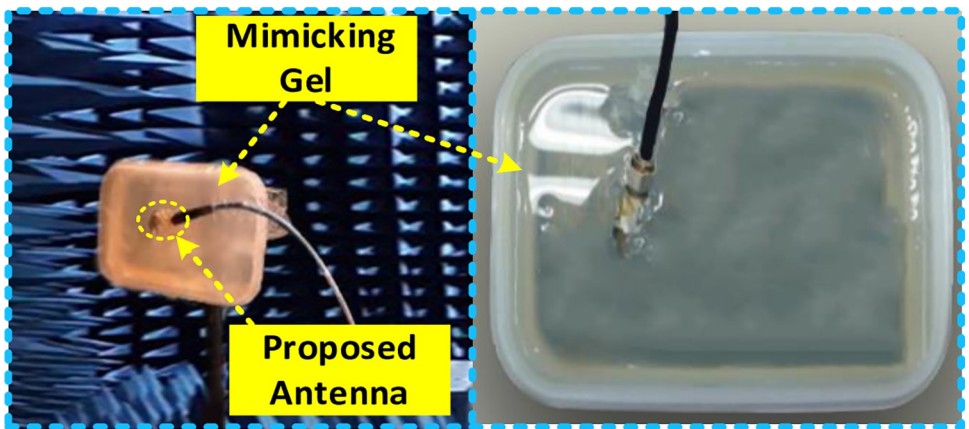

**Fig 16. In human tissue' skin-mimicking gel,' antenna measurements are performed.**

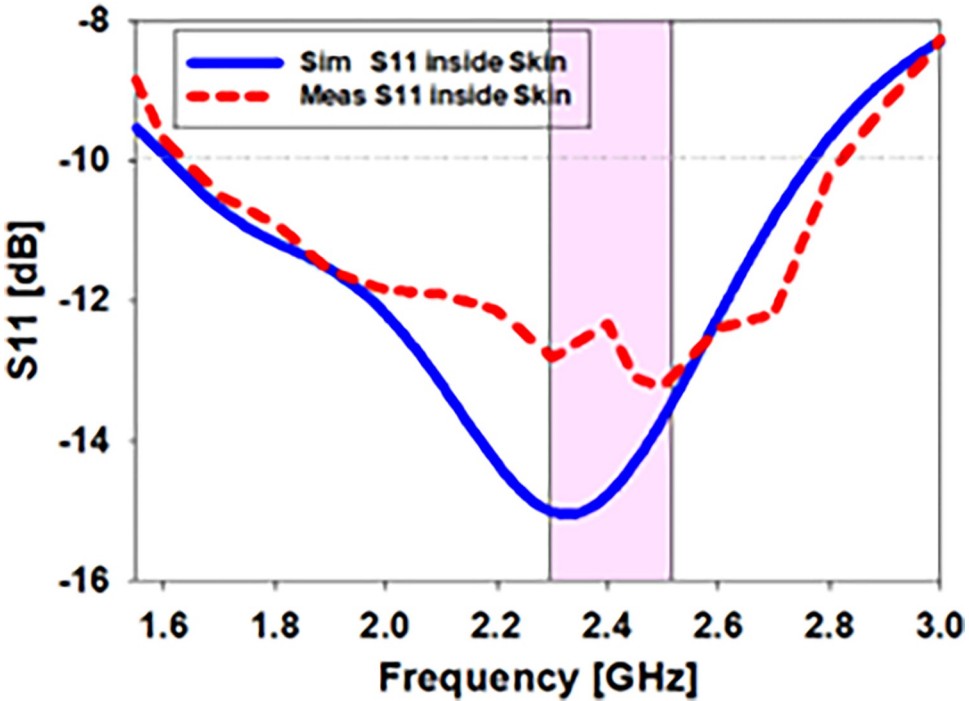

**Fig 17. Analyses of the antenna S11 in skin tissue compared to synthetic S11.**

antenna inside imitating gel had bandwidths ranging from 1.63 to 2.8 GHz (47.7%). These findings demonstrate a high degree of concordance between experimental and computer-simulated data.

An E and H-plane depicting the radiation pattern within skin tissue is shown in Fig 18. Antenna behavior in the skin is shown graphically in a 2D radiation pattern. As a consequence of these far field measurements, it is clear that the emission pattern of the antenna within skin is broadsided along E and narrower on H at 2.45 GHz. An in-side skin gain value of -13.8 dB

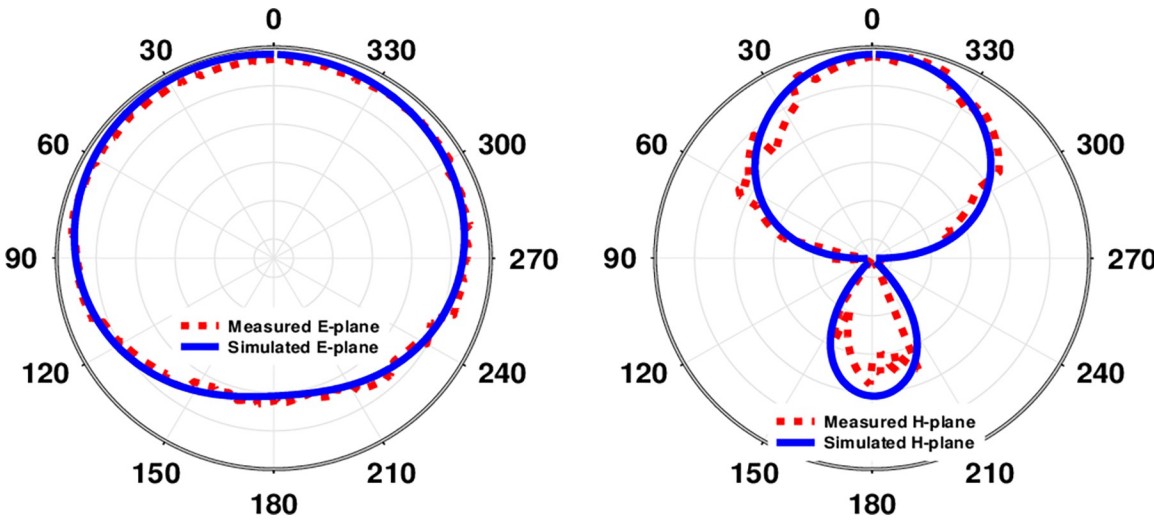

**Fig 18. Skin tissue was used to simulate and measure farfield outcomes at a frequency of 2.45 GHz.**

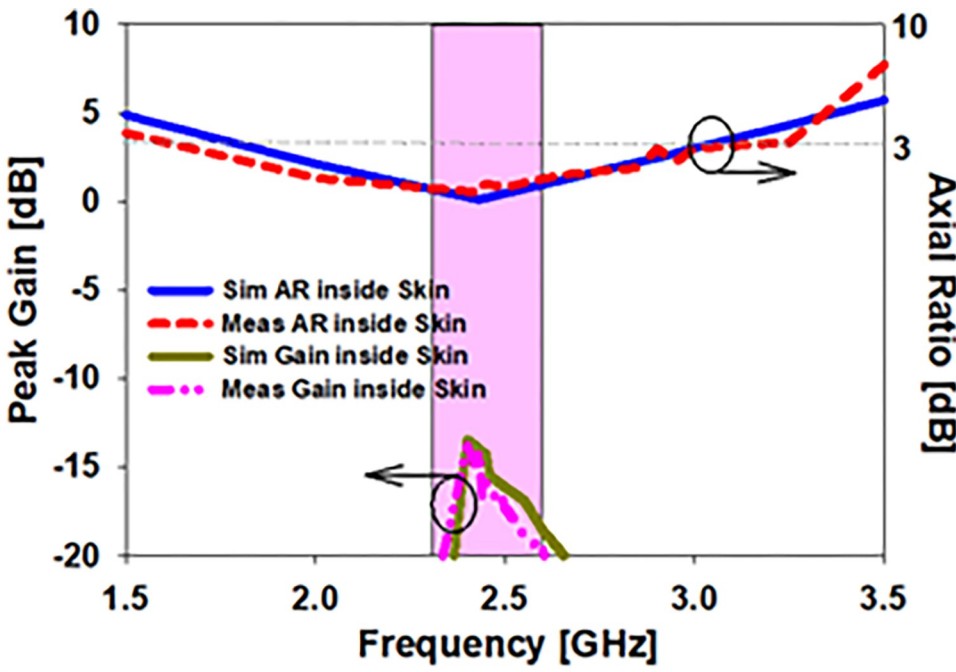

**Fig 19. The axial ratio of the prototype printed antenna within skin is compared to the simulation and test gains.**

was discovered to be simulated, whereas realized skin gain of -15.3 dB was found to be measured. Fig 19 depicts the simulated and observed axial ratio and realized gain, while Table 4 provides the corresponding results. Axial ratio bandwidths of 1.75 GHz to 3.1 GHz (55.1 percent) at 2.45 GHz were covered by the simulated antenna embedded in skin tissue, although axial ratio bandwidths of 1.68 GHz to 3 GHz (53.8 percent) in skin-mimicking gel were used.

## Specific absorption rate (sar)

Electromagnetic radiation may pose health concerns to humans, and these risks are quantified using the SAR. The following equation describes the link between input power and SAR [10]:

$$SAR = \frac{\sigma |E^2|}{\rho} \tag{5}$$

where '$\sigma$ and '$\rho$' the thermal conductivity (S/m), mass density (kg/m$^3$), and electric field intensity (V/m) are denoted by S and E, respectively. According to below [13], the electric power intensity is inversely proportional to the signal power.

$$Power \ (W/m^2) = \frac{(E(V/m))^2}{377} \tag{6}$$

It was possible to conduct SAR simulations if the antenna was retained within the skin. An

**Table 4. At a frequency of 2.45 GHz, both theoretical and experimental findings were compared.**

| Simulated inside Skin | | | Measured inside Skin | | |
|---|---|---|---|---|---|
| Gain (dB) | Imp. BW (%) | A.R BW (%) | Gain (DB) | Imp. B.W (%) | A.R B.W (%) |
| -13.8 | 46.9 | 55.1 | -15.3 | 47.7 | 53.8 |

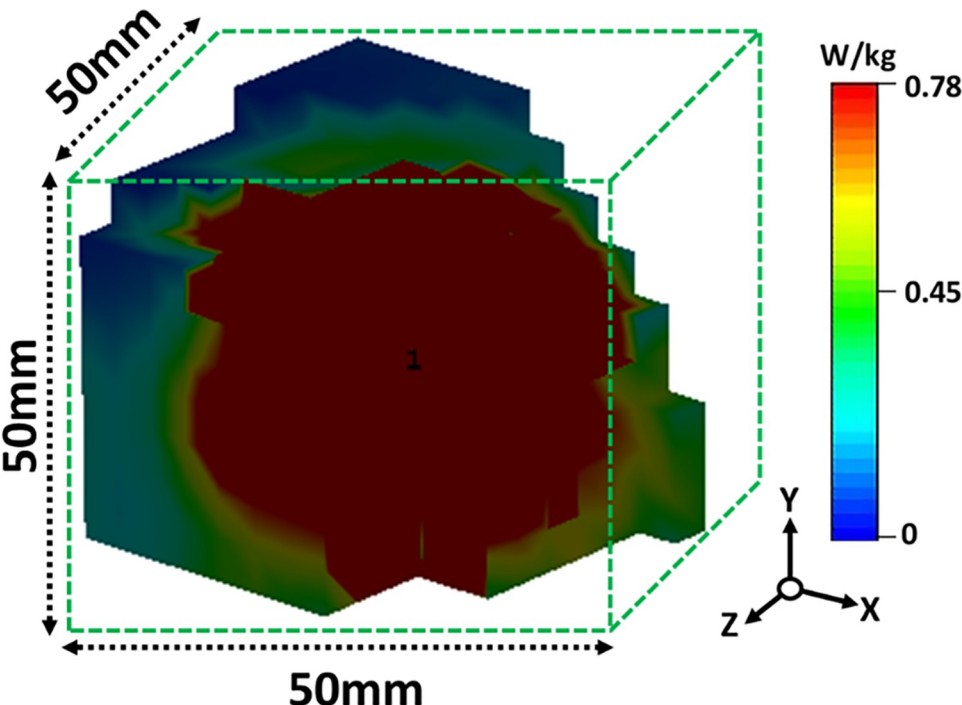

**Fig 20. At 2.45 GHz, the proposed antenna's SAR distribution within skin tissue measures 1 g.**

average of 1g of mass tissue is used to determine a device's SAR based on the IEEE/IEC 6270–1 standard. Fig 20 shows the SAR values estimated at 2.45 GHz within skin tissue at 0.78 W/kg.

## Conclusion

Within the scope of this work, a low-profile ISM band printed monopole antenna for use in biomedical implants is provided. The CST computer modelling application was utilized in the process of creating the provided antenna. This antenna is intended to function in the ISM Band, which operates at 2.45GHz. The total dimensions of the antenna are 21 millimeters by 13.5 millimeters by 0.254 millimeters, and it is constructed out of a flexible RT/Duroid 5880 material ($\varepsilon_r$ = 2.2, tan$\delta$ = 0.0009). The observed gain of the antenna is calculated to be -15.8 dB, despite the fact that the measured bandwidth of the antenna is reported to be 47.7 percent within skin tissue, respectively, and that the axial ratio bandwidths have been experimentally estimated as 53.8 percent inside skin imitating gel. The SAR values in skin tissue are lower than the FCC requirement of 0.78 W/kg. The development of medical applications has also been linked to issues of biocompatibility and safety. This antenna is small enough for in-body biomedical applications, making it an excellent candidate. An improved impedance bandwidth is another something we want to work on in order to achieve both an axial ratio and imped-ance bandwidth that are near enough to each other.

## Author Contributions

**Conceptualization:** Sarosh Ahmad, Ahmed Jamal Abdullah Al-Gburi.

**Data curation:** Sarosh Ahmad.

**Formal analysis:** Jalal Khan, Ahmed Jamal Abdullah Al-Gburi.

**Funding acquisition:** Mousa Hussein.

**Investigation:** Arslan Dawood Butt.

**Methodology:** Jalal Khan, Sarosh Ahmad, Ahmed Jamal Abdullah Al-Gburi.

**Project administration:** Arslan Dawood Butt.

**Resources:** Arslan Dawood Butt, Adnan Ghaffar.

**Software:** Adnan Ghaffar.

**Supervision:** Mousa Hussein.

**Validation:** Jalal Khan.

**Visualization:** Adnan Ghaffar.

**Writing – original draft:** Sarosh Ahmad.

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
