## [Decision Letter · Decision Letter 0]

23 Aug 2022

PONE-D-22-20756Single-Layered Compact Broadband Circularly Polarized Implantable Antenna for Medical ApplicationsPLOS ONE

Dear Dr. Ahmmad,

Thank you for submitting your manuscript to PLOS ONE. After careful consideration, we feel that it has merit but does not fully meet PLOS ONE’s publication criteria as it currently stands. Therefore, we invite you to submit a revised version of the manuscript that addresses the points raised during the review process.

ACADEMIC EDITOR:Authors should revise the manuscript addressing all the issues raised by reviewers. Note that you do not need to cite any suggested reference articles by the reviewer(s) if you find them not-related to your work.

We look forward to receiving your revised manuscript.

Kind regards,

Muhammad Zubair

Academic Editor

PLOS ONE

Journal Requirements:

"NO"

Please include your amended statements within your cover letter; we will change the online submission form on your 

Additional Editor Comments:

Authors should revise the manuscript addressing all the issues raised by reviewers. Note that you do not need to cite any suggested reference articles by the reviewer(s) if you find them not-related to your work.

Reviewers' comments:

Reviewer's Responses to Questions

**Comments to the Author**

1. Is the manuscript technically sound, and do the data support the conclusions?

Reviewer #1: Yes

Reviewer #2: Yes

2. Has the statistical analysis been performed appropriately and rigorously? 

Reviewer #1: Yes

Reviewer #2: Yes

3. Have the authors made all data underlying the findings in their manuscript fully available?

Reviewer #1: Yes

Reviewer #2: Yes

4. Is the manuscript presented in an intelligible fashion and written in standard English?

Reviewer #1: Yes

Reviewer #2: Yes

5. Review Comments to the Author

Reviewer #1: This article presents a single-layered compact broadband circularly polarized implantable antenna for medical applications. This antenna is intended to function in the ISM Band, which operates at 2.45GHz. Promising results have been achieved and well discussed in the well-organized manuscript. So, the results have been experimentally validated, and highlighted by providing a fair comparison with state-of-the-art. Although the concept and idea of this work were found very interesting and they seem attractive for the antenna and propagation society, authors are encouraged to address the following comments prior to final recommendation.

1) How can the bandwidth of the antenna be further improved?

2) What is the advantage of having dual functionality in free space and in human tissue?

3) Some important Abbreviations should be fully defined.

4) To design the antenna structures for implantable applications, the compact size is very important, therefore, authors are requested to mention some miniaturization methods in the introduction section along with proper references. Below are some helpful suggestions.

"A Comprehensive Survey of "Metamaterial Transmission-Line Based Antennas: Design, Challenges, and Applications"", IEEE Access, vol. 8, pp. 144778-144808, 2020.

“Miniaturized Planar-Patch Antenna Based on Metamaterial L-shaped Unit-Cells for Broadband Portable Microwave Devices and Multiband Wireless Communication Systems" IET Microwaves, Antennas & Propagation, Volume 12, Issue 7, 13 June 2018, p. 1080 – 1086.

“Compact Single Layer Travelling-Wave Antenna Design Using Metamaterial Transmission-Lines" Radio Science, Volume 52, Issue 12 December 2017, Pages 1510–1521.

“Periodic Array of Complementary Artificial Magnetic Conductor Metamaterials-Based Multiband Antennas for Broadband Wireless Transceivers” IET Microwaves, Antennas & Propagation, Volume 10, Issue 15, 10 December 2016, p. 1682 – 1691.

“Miniature CRLH-based ultra wideband antenna with gain enhancement for wireless communication applications”, ICT Express, Volume 2, Issue 2, June 2016, Pages 75–79.

"New Compact Antenna Based on Simplified CRLH-TL for UWB Wireless Communication Systems", International Journal of RF and Microwave Computer-Aided Engineering, Volume 26, Issue 3, March 2016, pages: 217–225.

“Metamaterial-Based Antennas for Integration in UWB Transceivers and Portable Microwave Handsets” International Journal of RF and Microwave Computer-Aided Engineering, Volume 26, Issue 1,January 2016, pages: 88–96.

"New Compact Printed Leaky-Wave Antenna with Beam Steering", Microwave and Optical Technology Letters, Volume 58, Issue 1, January 2016, Pages: 215–217.

"UWB Antenna Based on SCRLH-TLs for Portable Wireless Devices", Microwave and Optical Technology Letters, Volume 58, Issue 1, January 2016, Pages: 69–71.

"Composite Right-Left-Handed-Based Antenna with Wide Applications in Very-High Frequency-Ultra-High Frequency Bands for Radio Transceivers" IET Microwaves, Antennas & Propagation, Volume 9, Issue 15, 10 December 2015, p. 1713 – 1726.

“Bandwidth and Radiation Specifications Enhancement of Monopole Antennas Loaded with Split Ring Resonators” IET Microwaves, Antennas & Propagation, Volume 9, Issue 14, 19 November 2015, p. 1487 – 1496.

“The Resonating MTM Based Miniaturized Antennas for Wide-band RF-Microwave Systems”Microwave and Optical Technology Letters, Volume 57, Issue 10, pages 2339–2344, October 2015.

“Novel UWB Miniaturized Integrated Antenna Based on CRLH Metamaterial Transmission Lines” AEUE Elsevier- International Journal of Electronics and Communications, Volume 69, Issue 8, August 2015, Pages 1143–1149.

“Compact Antenna based on a Composite Right/Left Handed Transmission Line” Microwave and Optical Technology Letters, Volume 57, Issue 8, pages 1785–1788, August 2015.

“Printed planar patch antennas based on metamaterial”, International Journal of Electronics Letters, Volume 2, Issue 1, Jan 2014, pp 37-42.

“Design and Modeling of New UWB Metamaterial Planar Cavity Antennas with Shrinking of the Physical Size for Modern Transceivers,” International Journal of Antennas and Propagation, vol. 2013, Article ID 562538, 12 pages, 2013. doi:10.1155/2013/562538.

4) Some grammatical mistakes should be incorporated.

5) All figures should be properly cited in the main text.

6) The quality of the figure should be further improved.

7) Authors need to highlight the novelty of the proposed work.

8) Some old references should be replaced with the latest ones.

9) Reference part can be improved as per above mentioned suggestions.

Reviewer #2: In the introduction chapter, I suggest to introduce some more text and a few more references related to the actual practical applications (existing or future) of the proposed antenna solution. In other words to further elaborate on a few specific biomedical cases where the proposed antenna can be used. Also, add some further explanation about how and where in the human body the proposed antenna is implanted. Otherwise, I recommend the paper for publication.

6. PLOS authors have the option to publish the peer review history of their article (what does this mean?). If published, this will include your full peer review and any attached files.

Reviewer #1: No

Reviewer #2: No

---

## [Author Response · Author response to Decision Letter 0]

18 Nov 2022

PONE-D-22-20756

Single-Layered Compact Broadband Circularly Polarized Implantable Antenna for Medical Applications

PLOS ONE

………………………………………………………………………………………...

Reviewer Comments to the Author

Reviewer 1

This article presents a single-layered compact broadband circularly polarized implantable antenna for medical applications. This antenna is intended to function in the ISM Band, which operates at 2.45GHz. Promising results have been achieved and well discussed in the well-organized manuscript. So, the results have been experimentally validated, and highlighted by providing a fair comparison with state-of-the-art. Although the concept and idea of this work were found very interesting and they seem attractive for the antenna and propagation society, authors are encouraged to address the following comments prior to final recommendation.

Comment_1: How can the bandwidth of the antenna be further improved?

Response_1: We are thankful to the reviewer for this question. The bandwidth of the antenna can be further improved by varying the slots and by increasing the size of the antenna. Since, for the applications of biomedical implants, it is necessary to keep the size of the antenna as compact as possible.

……………………………

Comment_2: What is the advantage of having dual functionality in free space and in human tissue?

Response_2: The advantage of having dual functionality of the antenna both in free space and in human tissue is that there is no any need to design an external antenna to communicate with the antenna inside human body. The antenna will work for both at the same time. That will reduce the cost, time and efforts.

……………………………

Comment_3: Some important Abbreviations should be fully defined.

Response_3: We are thankful to the reviewer for pointing out the concerns related to the abbreviations. It is done in the revised manuscript.

……………………………

Comment_4: To design the antenna structures for implantable applications, the compact size is very important, therefore, authors are requested to mention some miniaturization methods in the introduction section along with proper references. Below are some helpful suggestions.

"A Comprehensive Survey of "Metamaterial Transmission-Line Based Antennas: Design, Challenges, and Applications"", IEEE Access, vol. 8, pp. 144778-144808, 2020.

“Miniaturized Planar-Patch Antenna Based on Metamaterial L-shaped Unit-Cells for Broadband Portable Microwave Devices and Multiband Wireless Communication Systems" IET Microwaves, Antennas & Propagation, Volume 12, Issue 7, 13 June 2018, p. 1080 – 1086.

“Compact Single Layer Travelling-Wave Antenna Design Using Metamaterial Transmission-Lines" Radio Science, Volume 52, Issue 12 December 2017, Pages 1510–1521.

“Periodic Array of Complementary Artificial Magnetic Conductor Metamaterials-Based Multiband Antennas for Broadband Wireless Transceivers” IET Microwaves, Antennas & Propagation, Volume 10, Issue 15, 10 December 2016, p. 1682 – 1691.

“Miniature CRLH-based ultra wideband antenna with gain enhancement for wireless communication applications”, ICT Express, Volume 2, Issue 2, June 2016, Pages 75–79.

"New Compact Antenna Based on Simplified CRLH-TL for UWB Wireless Communication Systems", International Journal of RF and Microwave Computer-Aided Engineering, Volume 26, Issue 3, March 2016, pages: 217–225.

“Metamaterial-Based Antennas for Integration in UWB Transceivers and Portable Microwave Handsets” International Journal of RF and Microwave Computer-Aided Engineering, Volume 26, Issue 1,January 2016, pages: 88–96.

"New Compact Printed Leaky-Wave Antenna with Beam Steering", Microwave and Optical Technology Letters, Volume 58, Issue 1, January 2016, Pages: 215–217.

"UWB Antenna Based on SCRLH-TLs for Portable Wireless Devices", Microwave and Optical Technology Letters, Volume 58, Issue 1, January 2016, Pages: 69–71.

"Composite Right-Left-Handed-Based Antenna with Wide Applications in Very-High Frequency-Ultra-High Frequency Bands for Radio Transceivers" IET Microwaves, Antennas & Propagation, Volume 9, Issue 15, 10 December 2015, p. 1713 – 1726.

“Bandwidth and Radiation Specifications Enhancement of Monopole Antennas Loaded with Split Ring Resonators” IET Microwaves, Antennas & Propagation, Volume 9, Issue 14, 19 November 2015, p. 1487 – 1496.

“The Resonating MTM Based Miniaturized Antennas for Wide-band RF-Microwave Systems «Microwave and Optical Technology Letters, Volume 57, Issue 10, pages 2339–2344, October 2015.

“Novel UWB Miniaturized Integrated Antenna Based on CRLH Metamaterial Transmission Lines” AEUE Elsevier- International Journal of Electronics and Communications, Volume 69, Issue 8, August 2015, Pages 1143–1149.

“Compact Antenna based on a Composite Right/Left Handed Transmission Line” Microwave and Optical Technology Letters, Volume 57, Issue 8, pages 1785–1788, August 2015.

“Printed planar patch antennas based on metamaterial”, International Journal of Electronics Letters, Volume 2, Issue 1, Jan 2014, pp 37-42.

“Design and Modeling of New UWB Metamaterial Planar Cavity Antennas with Shrinking of the Physical Size for Modern Transceivers,” International Journal of Antennas and Propagation, vol. 2013, Article ID 562538, 12 pages, 2013. doi:10.1155/2013/562538.

Response_4: We are thankful to the reviewer for these useful suggestions. The above references are cited in the revised manuscript. 

……………………………

Comment_5: Some grammatical mistakes should be incorporated.

Response_5: The grammatical errors are incorporated in the revised manuscript.

……………………………

Comment_6: All figures should be properly cited in the main text.

Response_6: Authors are thankful to the reviewer for this suggestion. The comment has been addressed in the revised manuscript.

……………………………

Comment_7: The quality of the figure should be further improved.

Response_7: We are thankful to the reviewer for this suggestion. The comment has been addressed in the revised manuscript.

……………………………

Comment_8: Authors need to highlight the novelty of the proposed work.

Response_8: The comment has been addressed in the revised manuscript. We have highlighted the novelty of the proposed work.

……………………………

Comment_9: Some old references should be replaced with the latest ones.

Response_9: The old references are replaced with the latest ones.

……………………………

Comment_10: Reference part can be improved as per above mentioned suggestions.

Response_10: Authors are thankful for this worthy suggestion.

………………………………………………………………………………………………

Reviewer 2

In the introduction chapter, I suggest to introduce some more text and a few more references related to the actual practical applications (existing or future) of the proposed antenna solution. In other words, to further elaborate on a few specific biomedical cases where the proposed antenna can be used. Also, add some further explanation about how and where in the human body the proposed antenna is implanted. Otherwise, I recommend the paper for publication.

Response: Authors are very thankful to the reviewer for this useful comment. We have incorporated new references in the revised manuscript. Since, from the title as well as the related applications, it is clear that the antenna is being utilized for biomedical implants for in-body communication. The proposed antenna can be utilized in several tissues of the human body such as the skin, fat or muscle. In our case, the antenna is being implemented in Skin tissue. Also, we have highlighted the section where we have given the description of the antenna use inside skin tissue. We have added and highlighted the explanation in the revised manuscript.

………………………………………………………………………………………………

---

## [Decision Letter · Decision Letter 1]

20 Dec 2022

Single-Layered Compact Broadband Circularly Polarized Implantable Antenna for Medical Applications

PONE-D-22-20756R1

Dear Dr. Hussein,

We’re pleased to inform you that your manuscript has been judged scientifically suitable for publication and will be formally accepted for publication once it meets all outstanding technical requirements.

Kind regards,

Muhammad Zubair

Academic Editor

PLOS ONE

Additional Editor Comments (optional):

The authors have addressed the concerns raised by the reviewers. The manuscript can be editorially accepted for publication.

Reviewers' comments:

Reviewer's Responses to Questions

**Comments to the Author**

1. If the authors have adequately addressed your comments raised in a previous round of review and you feel that this manuscript is now acceptable for publication, you may indicate that here to bypass the “Comments to the Author” section, enter your conflict of interest statement in the “Confidential to Editor” section, and submit your "Accept" recommendation.

Reviewer #1: All comments have been addressed

Reviewer #2: (No Response)

2. Is the manuscript technically sound, and do the data support the conclusions?

Reviewer #1: Yes

Reviewer #2: Yes

3. Has the statistical analysis been performed appropriately and rigorously? 

Reviewer #1: Yes

Reviewer #2: Yes

4. Have the authors made all data underlying the findings in their manuscript fully available?

Reviewer #1: Yes

Reviewer #2: Yes

5. Is the manuscript presented in an intelligible fashion and written in standard English?

Reviewer #1: Yes

Reviewer #2: Yes

6. Review Comments to the Author

Reviewer #1: Authors have successfully addressed the reviewer's concerns. So, looking at the quality of the revised manuscript which shows a significant improvement than its initial version, there are no more technical comments.

Reviewer #2: The authors have, in my opinion, addressed the issues raised by the reviewers, such that I can recommend the paper for publication in PLOS ONE.

7. PLOS authors have the option to publish the peer review history of their article (what does this mean?). If published, this will include your full peer review and any attached files.

Reviewer #1: No

Reviewer #2: No

---

## [Editor Report · Acceptance letter]

18 Jan 2023

PONE-D-22-20756R1 

Single-Fed Broadband CPW-Fed Circularly Polarized Implantable Antenna for Sensing Medical Applications 

Dear Dr. Hussein:

I'm pleased to inform you that your manuscript has been deemed suitable for publication in PLOS ONE. Congratulations! Your manuscript is now with our production department. 

Kind regards, 

on behalf of

Dr. Muhammad Zubair 

Academic Editor

PLOS ONE